# Synthesis, Crystal Structures, and Magnetic Properties of Mixed-Valent Tetranuclear Complexes with Y-Shaped $Mn^{II}_2Mn^{III}_2$ Core

**Masahiro Mikuriya** [1,*]**, Satoshi Kurahashi** [1]**, Seiki Tomohara** [1]**, Yoshiki Koyama** [1]**, Daisuke Yoshioka** [1]**, Ryoji Mitsuhashi** [1] **and Hiroshi Sakiyama** [2]

[1] Department of Applied Chemistry for Environment and Reserach Center for Coordination Molecule-Based Devices, School of Science and Technology, Kwansei Gakuin University, 2-1 Gakuen, Sanda 669-1337, Japan; kurahashi@kawakami-paint.co.jp (S.K.); ayg83493@gmail.com (S.T.); rakuraku0194@gmail.com (Y.K.); yoshi0431@gmail.com (D.Y.); mitsuhashi@kwansei.ac.jp (R.M.)

[2] Department of Science, Faculty of Science, Yamagata University, 1-4-12 Kojirakawa, Yamagata 990-8560, Japan; saki@sci.kj.yamagata-u.ac.jp

* Correspondence: junpei@kwansei.ac.jp; Tel.: +81-79-565-8365

**Abstract:** Tetranuclear $Mn^{II}_2Mn^{III}_2$ complexes with 1,3-bis(5-bromo-3-metoxysalicylidenaminomethyl)-2-propanol ($H_3$bmsap) and 1,3-bis(5-chloro-3-methoxysalicylidenaminomethyl)-2-propanol ($H_3$cmsap), [$Mn_4$(bmsap)$_2$(CH$_3$CO$_2$)$_3$(CH$_3$O)] (**3**) and [$Mn_4$(cmsap)$_2$(CH$_3$CO$_2$)$_3$(CH$_3$O)] (**4**), were synthesized and characterized by elemental analysis, infrared and diffused reflectance spectra and variable-temperature magnetic susceptibility measurements in the 2–300 K range. The crystal structures of **3** and **4** revealed a Y-shaped tetranuclear manganese cluster formed by the two Schiff-base ligands, three kinds of acetato ligands (bidentate, syn–anti-bridging, and syn–syn-bridging), and μ-methoxido ligand. The magnetic data showed the magnetic interactions among the four manganese atoms are antiferromagnetic as a whole within the tetranuclear cluster.

**Keywords:** manganese complex; tetranuclear complex; mixed-valent complex; antiferromagnetic interaction; spin coupling

## 1. Introduction

Within the search for new molecule-based devices, manganese complexes are interesting from the viewpoint of both bioinorganic chemistry and material sciences [1–8]. Specifically, tetranuclear manganese complexes have received considerable interest because the unique tetranuclear manganese cluster structure with calcium ion was found in the PS-II in green plants [9–12]. Many kinds of tetranuclear $Mn^{II}_2Mn^{III}_2$ complexes have appeared in the literature, and butterfly-shaped tetranuclear $Mn^{II}_2Mn^{III}_2$ complexes have remarkable magnetic properties with ferromagnetic couplings and SMM behavior [13–15]. Based on this background, we have focused on the synthesis of manganese complexes with multidentate organic ligands [16–28]. We and other researchers have studied coordination compounds based on a multidentate Schiff-base, 1,3-bis(salicylideneamino)-2-propanol, as a dinucleating ligand to make mononuclear, dinuclear, tetranuclear, and polynuclear manganese complexes [16,29–32]. Recently, we reported that tetranuclear manganese complexes with a Y-shaped core can be formed by the reaction of a multidentate Schiff-base ligand which has a methoxy group as a potential coordinating donor atom, 1,3-bis(3-methoxysalicylideneamino)-2-propanol ($H_3$msap) [28]. The reaction with manganese(II) acetate and manganese(II) benzoate afforded tetranuclear complexes, [$Mn_4$(msap)$_2$(CH$_3$CO$_2$)$_3$(CH$_3$O)(H$_2$O)] (**1**) and [$Mn_4$(msap)$_2$(C$_6$H$_5$CO$_2$)$_3$(CH$_3$O)] (**2**).

This is an interesting feature of tetranuclear manganese complexes related to the PS-II model. In order to increase the number of examples of this type of complexes, we introduced chloro- and bromo-groups to the 3-methoxysalicylidenamino moieties of H₃masp ligand to make new Schiff-base ligands, 1,3-bis(5-bromo-3-methoxysalicylideneamino)-2-propanol (H₃bmsap) and 1,3-bis(5-chloro-3-methoxysalicylideneamino)-2-propanol (H₃cmsap) (Figure 1). These new ligands have only one electron-withdrawing Br or Cl group for each benzene ring and we expected only a small effect on the tetranuclear molecule, and thus we can extend the tetranuclear examples. In order to construct tetranuclear complex, we used a similar method to make manganese complexes as that for the acetate complex **1** [28] and isolated two new complexes, [Mn₄(bmsap)₂(CH₃CO₂)₃(CH₃O)] (**3**) and [Mn₄(cmsap)₂(CH₃CO₂)₃(CH₃O)] (**4**). We report here on the synthesis, magnetic properties, and crystal structures of these tetranuclear complexes.

**Figure 1.** Multidentate Schiff-base ligands, 1,3-bis(3-methoxysalicylideneamino)-2-propanol (H₃msap; X = H) and its 5-bromo and 5-chloro derivatives, 1,3-bis(5-bromo-3-methoxysalicylideneamino)-2-propanol (H₃bmsap; X = Br) and 1,3-bis(5-chloro-3-methoxysalicylideneamino)-2-propanol (H₃cmsap; X = Cl).

## 2. Results and Discussion

### 2.1. Synthesis of Tetranuclear Manganese Complexes

The tetranuclear manganese complexes, [Mn₄(bmsap)₂(CH₃CO₂)₃(CH₃O)] (**3**) and [Mn₄(cmsap)₂(CH₃CO₂)₃(CH₃O)] (**4**), were synthesized by a 1:2 molar reaction of the Schiff-base ligand and manganese(II) acetate according to a similar method for [Mn₄(msap)₂(CH₃CO₂)₃(CH₃O)(H₂O)]·H₂O (**1**) [28]. Analytical C, H, and N data of the obtained complexes agree with the formulation of tetranuclear manganese.

### 2.2. Infrared Spectra of Tetranuclear Manganese Complexes

Infrared spectra of the present complexes **3** and **4** are shown with those of the complexes **1** and **2** in Figure 2. The fingerprint features are similar to each other, suggesting a similarity of the molecular structures of these complexes. The present complexes showed a C=N stretching band characteristic to the Schiff base ligands at 1619 cm⁻¹, which is definitely shifted to the lower frequency side, compared with those of the free Schiff-base ligands, H₃bmsap (1624 cm⁻¹) and H₃cmsap (1640 cm⁻¹), suggesting the coordination of the imino-nitrogen atoms to metal atoms. The complexes also showed two COO stretching bands at 1587–1534 and 1460–1396 cm⁻¹ with the frequency difference characteristic of syn–syn-bridging, syn–anti-bridging, and bidentate carboxylato as similar to the tetranuclear complexes **1** and **2** [28]. The frequency separation between the antisymmetric and symmetric COO stretching bands can be deduced as $\Delta = \nu_{as}(COO) - \nu_s(COO)$: $1587 - 1396 = 191$ cm⁻¹, $1555 - 1417 = 138$ cm⁻¹, $1534 - 1449 = 85$ cm⁻¹ for **3**; $1579 - 1407 = 172$ cm⁻¹, $1545 - 1439 = 106$ cm⁻¹, $1534 - 1460 = 74$ cm⁻¹ for **4**., corresponding to those of syn–syn-bridging, syn–anti-bridging, and bidentate acetato ligands [33,34].

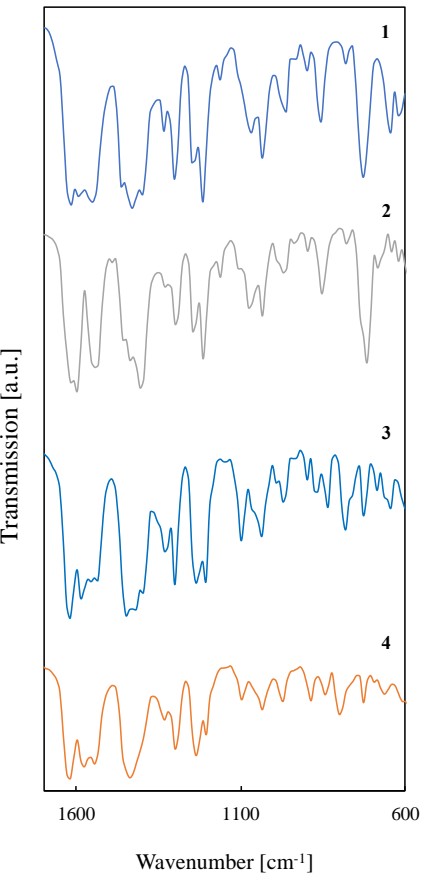

**Figure 2.** Infrared spectra of [Mn$_4$(bmsap)$_2$(CH$_3$CO$_2$)$_3$(CH$_3$O)] (**3**) and [Mn$_4$(cmsap)$_2$(CH$_3$CO$_2$)$_3$(CH$_3$O)] (**4**) compared with those of [Mn$_4$(msap)$_2$(CH$_3$CO$_2$)$_3$(CH$_3$O)(H$_2$O)]·H$_2$O (**1**) and [Mn$_4$(msap)$_2$(C$_6$H$_5$CO$_2$)$_3$(CH$_3$O)] (**2**).

### 2.3. Electronic Spectra of Tetranuclear Manganese Complexes

The diffused reflectance spectra of the present complexes (Figure S1) may be characterized as broad spectral features with some peaks in the UV-vis–NIR region, as previously reported for **1** and **2** [28]. The LMCT (Ligand-to-Metal Charge Transfer) bands may be responsible for the high-intensity absorptions in the UV region. In particular, the absorption band at the near-UV region (434sh nm for **3**; 442sh for **4**) can be assumed as a characteristic LMCT band of phenolato-oxygen to manganese(III) as found in some Schiff-base manganese(III) complexes [24,35]. Furthermore, d-d bands due to six- or five-coordinated manganese(III) ions appeared and hid behind the strong near-UV bands in the visible and NIR regions until around 1000 nm [35].

### 2.4. Crystal Structures of Tetranuclear Manganese Complexes

Single crystals suitable for X-ray diffraction work were obtained for complexes **3** and **4**, although the crystals of the latter complex were twinned. Crystal data and refinement parameters are given in Table 1. Selected bond lengths and angles are given in Tables 2 and 3. An ORTEP (Oak Ridge Thermal Ellipsoid Plot) drawing of the molecular structure of **3** is shown in Figure 3. One Schiff-base bmsap$^{3-}$ ligand binds the Mn1, Mn2, and Mn3 atoms by the O1, N1, and O3 atoms, the O4, N2, and O3 atoms, and the O3 atom, respectively, with a folding structure of the Schiff-base ligand. On the other hand, the other bmsap$^{3-}$ ligand binds the Mn1, Mn3, and Mn4 atoms by the O6, N3, and O8 atoms, the O8, N4, and O9 atoms, and the O9 and O10 atoms, respectively. The Mn1 atom is six-coordinated by the N$_2$O$_4$ donor set from the two Schiff-base ligands in an elongated octahedral geometry with the longer O1-Mn1-O3 axis. The Mn2 atom is five-coordinated by the NO$_4$ donor set from one Schiff-base, one methoxido-oxygen, and one acetato-oxygen atom in a distorted square-pyramidal

geometry, having the $\tau$ value of 0.217 around the Mn2 atom [36]. The Mn3 atom is six-coordinated by the $NO_5$ donor set from two Schiff-base, one methoxido-oxygen atom, and one acetato-oxygen atom in a distorted octahedral geometry with relatively long bond distances. The Mn4 atom is also six-coordinated by the $O_6$ donor set from one Schiff-base and four acetato-oxygen atoms in a distorted octahedron with longer Mn-O bonds. The charge balance and bond parameters around the manganese centers suggested the Mn1 and Mn2 atoms should be in a Mn(III) oxidation state, whereas the Mn3 and Mn4 atoms should be in a Mn(II) oxidation state. The bond valence sum calculation supported this mixed-valent formula [37,38]. The four manganese atoms are located at the corners of a Y-shaped core with distances between the four manganese atoms of 3.530(2) Å for Mn1···Mn2, 3.237(1) Å for Mn2···Mn3, 3.466(2) Å for Mn1···Mn3, and 3.494(2) Å for Mn3···Mn4, respectively, where the Mn1 and Mn2 atoms are bridged by the $\mu_3$-O3 alkoxido-oxygen atom of the Schiff-base ligand, the Mn1 and Mn3 atoms are bridged by the $\mu_3$-O3 alkoxido-oxygen atom and $\mu$-O8 alkoxido-oxygen atom of the Schiff-base ligand, the Mn2 and Mn3 atoms are bridged by the $\mu_3$-O3 alkoxido-oxygen atom and methoxido-oxygen O17 atom, the Mn2 and Mn4 atoms are bridged by the acetato ligand, and the Mn3 and Mn4 atoms are bridged by the phenoxido-oxygen O9 atom and acetato ligand. It is to be noted that three kinds of coordination modes of acetato ligands, syn–syn-bridging, syn–anti-bridging, and bidentate, are found in the same molecule. A similar feature was observed in the hexanuclear copper(II) complex with 1,4,8,11-tetrakis(3-methoxysalicylideneaminoethyl)-1,4,8,11-tetraazacycloteradecane ($H_4$tmsaec), $[Cu_6(CH_3CO_2)_8(tmsaec)]$, where monodentate, syn–syn-bridging, and bidentate modes were found for the acetato ligands [34]. The molecular structure is similar to that of **1**, but there is no coordinating water molecule nor monodentate acetato-ligand here. This structure is almost the same as that of **2**. A packing diagram of **3** is shown in Figure 4. The tetranuclear molecules are separated each other in the crystal.

**Table 1.** Crystallographic data and refinement parameters of 3 and 4.

|  | **3** | **4** |
|---|---|---|
| Empirical formula | $C_{45}H_{46}Br_4Mn_4N_4O_{17}$ | $C_{48.3}H_{54.2}Cl_4Mn_4N_7O_{18.3}$ |
| Formula weight | 1454.26 | 1359.12 |
| Temperature/K | 90 | 90 |
| Crystal dimensions/mm | $0.23 \times 0.12 \times 0.08$ | $0.08 \times 0.15 \times 0.35$ |
| Crystal system | monoclinic | triclinic |
| Space group | $C2/c$ | $P\bar{1}$ |
| $a/Å$ | 32.754(9) | 12.5259(14) |
| $b/Å$ | 15.121(4) | 15.5053(17) |
| $c/Å$ | 29.893(13) | 16.8716(18) |
| $\alpha/°$ | 90 | 102.278(2) |
| $\beta/°$ | 117.132(3) | 104.869(2) |
| $\gamma/°$ | 90 | 110.789(2) |
| $V/Å^3$ | 13176(8) | 2788.6(5) |
| $Z$ | 8 | 2 |
| $d_{calcd.}/g·cm^{-3}$ | 1.466 | 1.619 |
| $\mu/mm^{-1}$ | 3.234 | 1.153 |
| $F(000)$ | 5760 | 1387 |
| Reflections collected | 41795 | 12414 |
| Independent reflections | 16231 | 12414 |
| $\theta$ range for data collection | 1.397 to 28.715° | 2.569 to 27.496° |
| Data/restraints/parameters | 16231/0/675 | 12414/0/744 |
| $R1$, $wR2$ $[I > 2\sigma(I)]$ [a] | 0.0785, 0.1756 | 0.0725, 0.1432 |
| $R1$, $wR2$ (all data) [a] | 0.1798, 0.2135 | 0.1252, 0.1593 |
| Goodness-of-fit on $F^2$ | 0.976 | 0.979 |
| CCDC (The Cambridge Crystallographic Data Centren) number | 1887620 | 1887482 |

[a] $R1 = \Sigma||F_o| - |F_c||/\Sigma|F_o|$; $wR2 = [\Sigma w(F_o^2 - F_c^2)^2/\Sigma w(F_o^2)^2]^{1/2}$.

**Table 2.** Selected bond lengths (Å) and bond angles (°) for complex **3**.

| | $[Mn_4(bmsap)_2(CH_3CO_2)_3(CH_3O)]$ (3) | | |
|---|---|---|---|
| **Bond** | **Bond Length/Å** | **Bond** | **Bond Angle/°** |
| Mn1⋯Mn2 | 3.530(2) | Mn1-O3-Mn2 | 110.2(3) |
| Mn2⋯Mn3 | 3.237(1) | Mn2-O3-Mn3 | 99.9(2) |
| Mn3⋯Mn1 | 3.466(2) | Mn1-O3-Mn3 | 94.4(2) |
| Mn4⋯Mn3 | 3.494(2) | Mn2-O17-Mn3 | 105.1(2) |
| Mn1-O1 | 2.067(5) | Mn3-O9-Mn4 | 110.2(2) |
| Mn1-O3 | 2.399(4) | Mn1-O8-Mn3 | 113.0(2) |
| Mn1-O6 | 1.911(5) | - | - |
| Mn1-O8 | 1.889(5) | - | - |
| Mn1-N1 | 2.056(6) | - | - |
| Mn1-N3 | 1.993(5) | - | - |
| Mn2-O3 | 1.891(4) | - | - |
| Mn2-O4 | 1.881(4) | - | - |
| Mn2-O11 | 2.085(5) | - | - |
| Mn2-O17 | 1.872(5) | - | - |
| Mn2-N2 | 1.981(6) | - | - |
| Mn3-O3 | 2.324(4) | - | - |
| Mn3-O8 | 2.261(5) | - | - |
| Mn3-O9 | 2.162(5) | - | - |
| Mn3-O13 | 2.080(4) | - | - |
| Mn3-O17 | 2.196(4) | - | - |
| Mn3-N4 | 2.241(6) | - | - |
| Mn4-O9 | 2.098(5) | - | - |
| Mn4-O10 | 2.343(7) | - | - |
| Mn4-O12 | 2.123(5) | - | - |
| Mn4-O14 | 2.081(6) | - | - |
| Mn4-O15 | 2.310(6) | - | - |
| Mn4-O16 | 2.196(5) | - | - |

**Table 3.** Selected bond lengths (Å) and bond angles (°) for complex 4.

| | $[Mn_4(cmsap)_2(CH_3CO_2)_3(CH_3O)]$ (4) | | |
|---|---|---|---|
| **Bond** | **Bond Length/Å** | **Bond** | **Bond Angle/°** |
| Mn1⋯Mn2 | 3.590(1) | Mn1-O5-Mn2 | 113.8(2) |
| Mn2⋯Mn3 | 3.181(1) | Mn2-O5-Mn3 | 99.4(1) |
| Mn3⋯Mn1 | 3.391(1) | Mn1-O5-Mn3 | 93.3(1) |
| Mn4⋯Mn3 | 3.562(1) | Mn2-O6-Mn3 | 100.8(2) |
| Mn1-O1 | 2.041(3) | Mn3-O11-Mn4 | 113.0(2) |
| Mn1-O5 | 2.387(3) | Mn1-O7-Mn3 | 108.6(2) |
| Mn1-O7 | 1.892(3) | - | - |
| Mn1-O8 | 1.885(3) | - | - |
| Mn1-N1 | 2.064(4) | - | - |
| Mn1-N3 | 2.000(4) | - | - |
| Mn2-O4 | 1.860(3) | - | - |
| Mn2-O5 | 1.885(3) | - | - |
| Mn2-O6 | 1.877(3) | - | - |
| Mn2-O15 | 2.100(3) | - | - |
| Mn2-N2 | 1.975(4) | - | - |
| Mn3-O5 | 2.273(3) | - | - |
| Mn3-O6 | 2.240(3) | - | - |
| Mn3-O7 | 2.275(3) | - | - |
| Mn3-O11 | 2.125(3) | - | - |
| Mn3-O17 | 2.086(3) | - | - |
| Mn3-N4 | 2.210(4) | - | - |
| Mn4-O10 | 2.336(3) | - | - |
| Mn4-O11 | 2.148(3) | - | - |
| Mn4-O12 | 2.306(4) | - | - |
| Mn4-O13 | 2.262(4) | - | - |
| Mn4-O14 | 2.166(3) | - | - |
| Mn4-O16 | 2.094(4) | - | - |

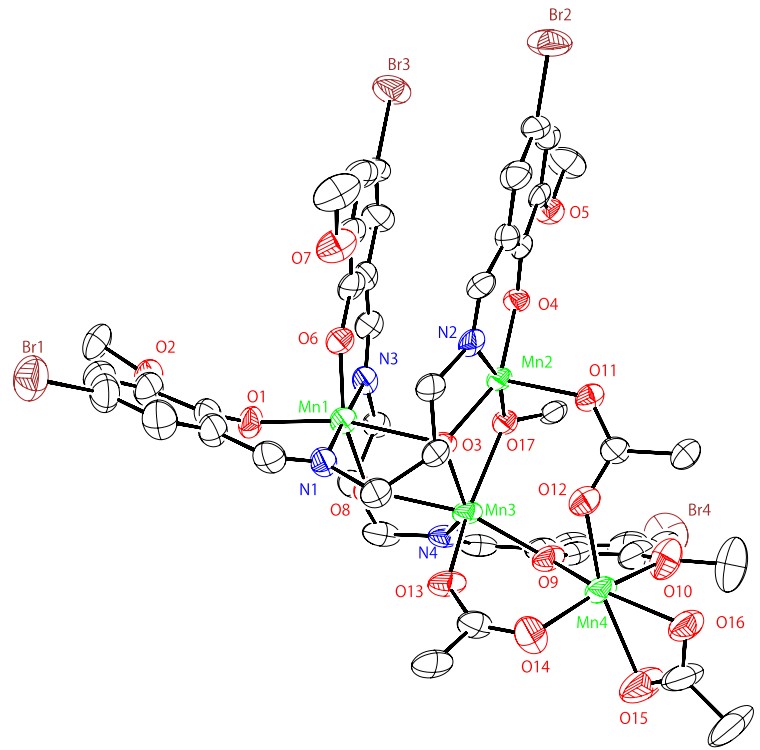

**Figure 3.** ORTEP (Oak Ridge Thermal Ellipsoid Plot) drawing of the molecular structure for [Mn$_4$(bmsap)$_2$(CH$_3$CO$_2$)$_3$(CH$_3$O)] (**3**).

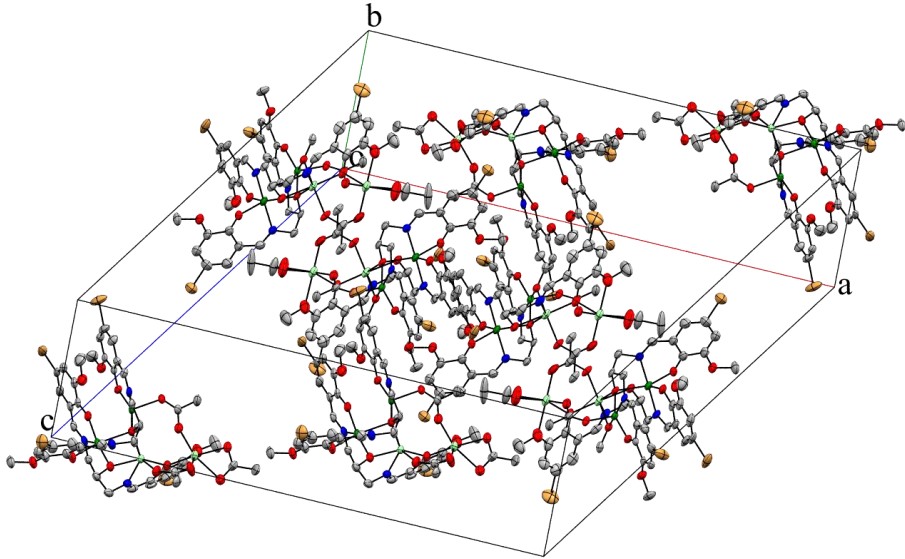

**Figure 4.** Packing diagram for [Mn$_4$(bmsap)$_2$(CH$_3$CO$_2$)$_3$(CH$_3$O)] (**3**).

An ORTEP drawing of the molecular structure of **4** is shown in Figure 5. The molecular structure is almost the same as that of **3**. The four manganese atoms occupied the four corners of a Y-shaped core with the distances of 3.590(1) Å for Mn1···Mn2, 3.181(1) Å for Mn2···Mn3, 3.391(1) Å for Mn1···Mn3, and 3.562(1) Å for Mn3···Mn4, respectively, where the Mn1 and Mn2 atoms are bridged by the μ$_3$-O5 alkoxido-oxygen atom of the Schiff-base ligand, the Mn1 and Mn3 atoms are bridged by the μ$_3$-O5 alkoxido-oxygen atom and μ-O7 alkoxido-oxygen atom of the Schiff-base ligand, the Mn2 and Mn3 atoms are bridged by the μ$_3$-O5 alkoxido-oxygen atom and methoxido-oxygen O6 atom, the Mn2 and Mn4 atoms are bridged by the syn–anti-acetato ligand, and the Mn3 and Mn4 atoms are bridged by

the phenoxido-oxygen O11 atom and syn–syn-acetato ligand. The packing diagram of **4** is depicted in Figure 6. We can see that the tetranuclear molecules are separated well in the crystal.

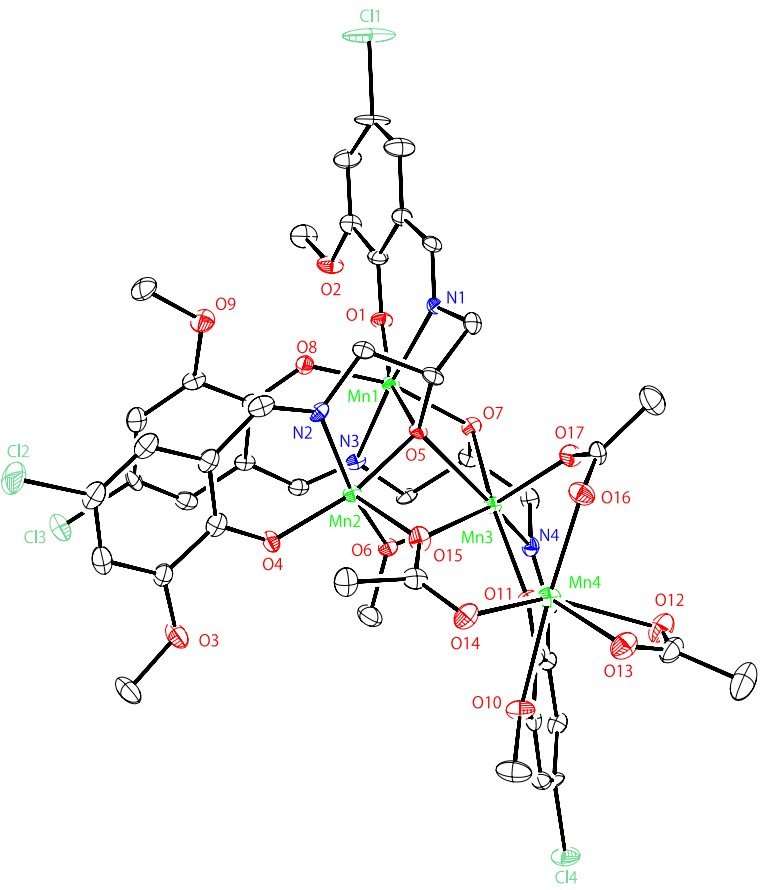

**Figure 5.** ORTEP drawing of the molecular structure for $[Mn_4(cmsap)_2(CH_3CO_2)_3(CH_3O)]$ (**4**).

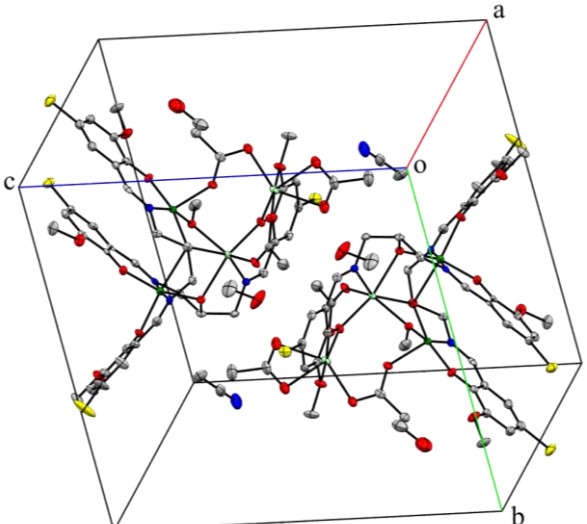

**Figure 6.** Packing diagram for $[Mn_4(cmsap)_2(CH_3CO_2)_3(CH_3O)]$ (**4**).

## 2.5. Magnetic Data of Tetranuclear Manganese Complexes

The magnetic properties for the tetranuclear manganese complexes **3** and **4** are displayed in Figure 7 as the temperature variations of $\chi_M T$ in the temperature range of 2–300 K. The effective

magnetic moments of **3** and **4** are 9.77 $\mu_B$ and 10.17 $\mu_B$, respectively, per $Mn^{II}_2Mn^{III}_2$ unit at 300 K, and a little lower than the theoretical value at room temperature. The calculated spin-only value is 10.86 $\mu_B$ for non-interacting two $S = 5/2$ spins and two $S = 2$ spins. When cooling, the magnetic moments of **3** and **4** steadily decrease from 300 K to around 100 K, and then abruptly diminish to a value of approximately 3.13 $\mu_B$ and 1.91 $\mu_B$ at 2 K, respectively. Overall, each tetranuclear manganese complex including **1** and **2** shows a similar pattern of magnetic moment decreasing with the lowering of temperature, suggesting that the magnetic coupling is antiferromagnetic as a whole. Therefore, the magnetic properties of the present complexes were described by the following model, taking account of the magnetic exchange interactions: $J_1$ ($Mn^{III}$-$Mn^{III}$ bridged by $\mu_3$-alkoxido-oxygen), $J_2$ ($Mn^{III}$-$Mn^{II}$ bridged by $\mu_3$-alkoxido-oxygen and $\mu$-methoxido-oxygen), $J_3$ ($Mn^{III}$-$Mn^{II}$ bridged by $\mu_3$-alkoxido-oxygen and $\mu$-alkoxido-oxygen), $J_4$ ($Mn^{II}$-$Mn^{II}$ bridged by $\mu$-phenoxido-oxygen and $\mu$-acetato), $J_5$ ($Mn^{III}$-$Mn^{II}$ bridged by $\mu$-acetato), and $J_6$ ($Mn^{III}$-$Mn^{II}$ without bridge) as shown in Figure 8. To determine the $J_1$, $J_2$, $J_3$, $J_4$, $J_5$, and $J_6$ values, the $\chi_M T$ versus $T$ data for **1**, **2**, **3**, and **4**, were fit to the theoretical expression based on the isotropic Heisenberg spin model given by Equation (1), using the program PHI [39].

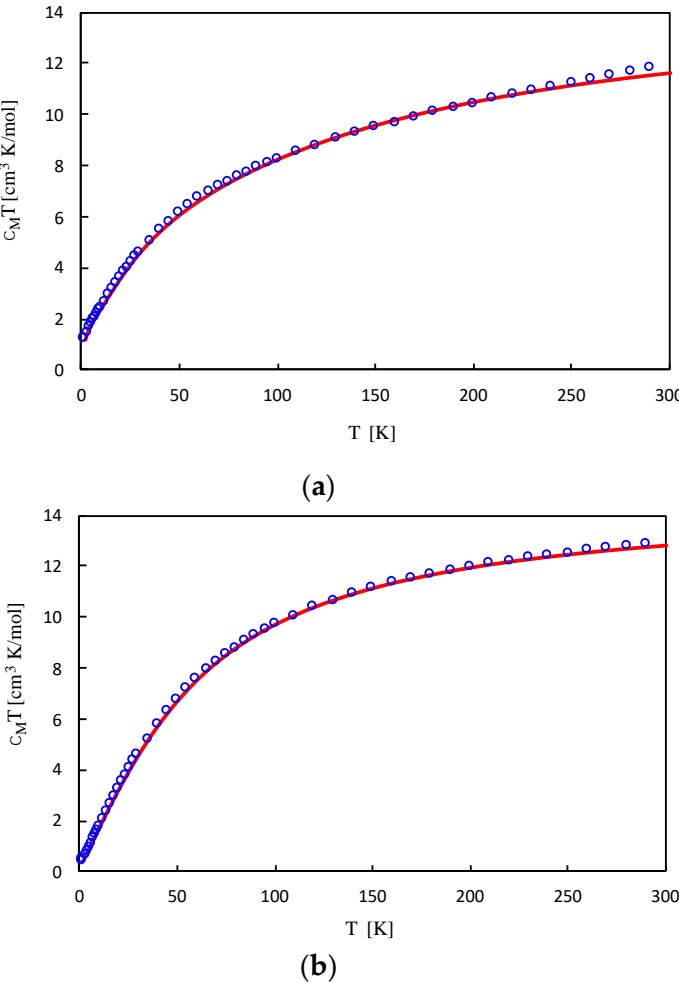

(**a**)

(**b**)

**Figure 7.** Temperature dependence of $\chi_M T$ (blue circles) and the fitting line (red curve) of tetranuclear complexes (**a**) [Mn$_4$(bmsap)$_2$(CH$_3$CO$_2$)$_3$(CH$_3$O)] (**3**) and (**b**) [Mn$_4$(cmsap)$_2$(CH$_3$CO$_2$)$_3$(CH$_3$O)] (**4**).

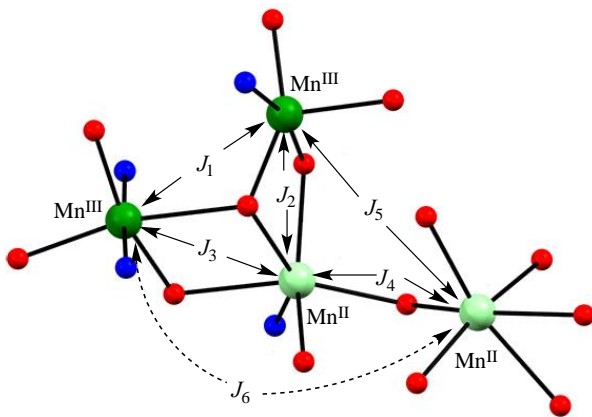

**Figure 8.** Magnetic coupling scheme in the present tetranuclear $Mn^{II}_2Mn^{III}_2$ complexes.

$$H = -2J_1S(Mn^{III}) \cdot S(Mn^{III}) - 2J_2S(Mn^{III}) \cdot S(Mn^{II}) - 2J_3S(Mn^{III}) \cdot S(Mn^{II}) -$$
$$2J_4S(Mn^{II}) \cdot S(Mn^{II}) - 2J_5S(Mn^{III}) \cdot S(Mn^{II}) - 2J_6S(Mn^{III}) \cdot S(Mn^{II}) \tag{1}$$

In order to avoid overparameterization, the *g* values of the four manganese atoms were fixed to be 2.00 and $J_6$ was set to be 0 cm$^{-1}$ because of there being no intervening bridging group between these Mn atoms. Good fits were obtained for all four complexes, and the results are shown as solid lines in Figure 7. The fitting parameters are listed in Table 4. All of the *J* values except for $J_5$ are negative, in accord with overall antiferromagnetic couplings in **1**, **2**, **3**, and **4**. Larger $-J_1$ values are understandable, because antiferromagnetic coupling between $Mn^{III}$ and $Mn^{III}$ is usually stronger than that between $Mn^{II}$ and $Mn^{III}$ or $Mn^{II}$ [17,19]. A ferromagnetic coupling was observed for $J_5$. This may come from magnetic interaction via the syn–anti-bridging acetato ligand.

**Table 4.** Fitting parameters for tetranuclar manganese complexes **1**–**4**.

|  | Complex 1 | Complex 2 | Complex 3 | Complex 4 |
|---|---|---|---|---|
| $J_1$ | −4.91 cm$^{-1}$ | −7.43 cm$^{-1}$ | −10.78 cm$^{-1}$ | −5.36 cm$^{-1}$ |
| $J_2$ | −1.77 cm$^{-1}$ | −2.56 cm$^{-1}$ | −7.01 cm$^{-1}$ | −2.30 cm$^{-1}$ |
| $J_3$ | −0.75 cm$^{-1}$ | −2.63 cm$^{-1}$ | −5.19 cm$^{-1}$ | −2.99 cm$^{-1}$ |
| $J_4$ | −2.20 cm$^{-1}$ | −2.07 cm$^{-1}$ | −2.14 cm$^{-1}$ | −3.29 cm$^{-1}$ |
| $J_5$ | 7.84 cm$^{-1}$ | 2.33 cm$^{-1}$ | 0.61 cm$^{-1}$ | 1.69 cm$^{-1}$ |
| $J_6$ | 0 cm$^{-1}$ | 0 cm$^{-1}$ | 0 cm$^{-1}$ | 0 cm$^{-1}$ |

## 3. Materials and Methods

All the chemicals were commercial products and were used as supplied. The Schiff-base ligands H$_3$bmsap and H$_3$cmsp were prepared according to a method reported for H$_3$msap in the literature [28].

### 3.1. Synthesis of 1,3-Bis(5-bromo-3-methoxysalicylideneamino)-2-propanol (H$_3$bmsap)

1,3-Diamino-2-propanol (100 mg, 0.1 mmol) and 5-bromo-3-methoxysalicylaldehyde (500 mg, 0.2 mmol) were dissolved in ethanol (30 cm$^3$). The mixture solution was refluxed for 3 h. The resulting yellow precipitate was filtered off and washed with ethanol. Yield: 460 mg (75%). Anal. found: C, 44.42; H, 3.99; N, 5.42%. Calcd for C$_{19}$H$_{20}$Br$_2$N$_2$O$_5$: C, 44.21; H, 3.91; N, 5.43%. IR (KBr): $v$(OH) 3219, $v$(C=N) 1624 cm$^{-1}$. $^1$H NMR (400 MHz, Chloroform-*d*): δ 13.61 (s, 2H, -OH), 8.42 (s, 2H, -N=CH-), 7.06–6.98 (m, 4H, aryl-H), 4.27 (m, 1H, -CH-), 3.90 (s, 6H, -CH$_3$), 3.91–3.87 (m, 2H, -CH$_2$-), 3.78–3.74 (m, 2H, -CH$_2$), 1.99 (s, 1H, -OH).

### 3.2. Synthesis of 1,3-Bis(5-chloro-3-methoxysalicylideneamino)-2-propanol ($H_3$cmsap)

$H_3$cmsap was prepared as for $H_3$bmsap using 5-chloro-3-methoxysalicylaldehyde in place of 5-bromo-3-methoxysalicylaldehyde.

### 3.3. Synthesis of [$Mn_4$(bmsap)$_2$($CH_3CO_2$)$_3$($CH_3O$)] (**3**)

To an acetonitrile solution (2 cm$^3$) of $H_3$bmsap (52 mg, 0.1 mmol), a mixed solution of manganese(II) acetate tetrahydrate (50 mg, 0.2 mmol) in acetonitrile (1 cm$^3$)—methanol (1 cm$^3$) solution was added, and then three drops of triethylamine were added to this solution. After the reaction mixture was stirred for 1h, the mixture was filtered. Diethyl ether was layered on the filtrate and left for several days at room temperature. The deposited crystals were filtered off and desiccated in vacuo. Yield: 33 mg, 44% (based on the Schiff-base ligand). Found C 35.73, H 3.32, N 3.71%. Calcd for $C_{45}H_{52}Br_4Mn_4N_4O_{20}$ (**3**·$3H_2O$): C 35.83, H 3.47, N 3.71%. IR (KBr): $v$(OH) 3329, $v$(C=N) 1619, $v_{as}$($CO_2^-$) 1587, 1555, 1534, $v_s$($CO_2^-$) 1449, 1417, 1396 cm$^{-1}$. Diffuse reflectance spectrum: $\lambda_{max}$ 236, 276, 370, 434sh, 540sh, 660sh, 840sh nm.

### 3.4. Synthesis of [$Mn_4$(cmsap)$_2$($CH_3CO_2$)$_3$($CH_3O$)] (**4**)

To an acetonitrile solution (2 cm$^3$) of $H_3$cmsap (22 mg, 0.05 mmol), a methanol solution (1 cm$^3$) of manganese(II) acetate tetrahydrate (24 mg, 0.1 mmol) was added, and then three drops of triethylamine were added to this solution. After the reaction mixture was stirred for 1 h, the mixture was filtrated. Diethyl ether was layered on the filtrate and left for several days at 5 °C. The deposited crystals were filtered off and desiccated in vacuo. Yield: 6 mg, 36% (based on the Schiff-base ligand). Found C 41.72, H 3.76, N 4.34%. Calcd for $C_{45}H_{48}Cl_4Mn_4N_4O_{18}$ (**4**·$H_2O$): C 41.75, H 3.74, N 4.33%. IR (KBr): $v$(OH) 3369, $v$(C=N) 1619, $v_{as}$($CO_2^-$) 1577, 1545, 1534sh, $v_s$($CO_2^-$) 1460sh, 1439, 1407sh cm$^{-1}$. Diffuse reflectance spectrum: $\lambda_{max}$ 302sh, 362, 386, 442sh, 550sh, 650sh, 850sh nm.

Analyses of C, H, and N of the ligands and complexes were carried out with a Thermo-Finnigan FLASH EA1112 series CHNO-S analyzer (Thermo-Finnigan, USA). Infrared spectra were recorded on KBr pellets in the range 4000–600 cm$^{-1}$, with a JASCO MFT-2000 FT-IR Spectrophotometer (JASCO, Japan). Diffused reflectance spectrum was taken with a Shimadzu UV-vis-NIR Spectrophotometer Model UV-3100 (Shimadzu, Japan) in the range 200–1500 nm. Variable-temperature magnetic data (2–300 K) were obtained using a Quantum Design MPMS-XL7 SQUID magnetometer (Quantum Design, USA).

The crystal data of **3** and **4** were collected with a Bruker CCD diffractometer Smart APEX (Bruker, USA) fitted with Mo Kα radiation and a graphite monochromator. The structure of **3** was solved by an intrinsic phasing method, and refined by full-matrix least-squares methods. The structure of **4** was solved by an intrinsic method as a 2-component twin with only the non-overlapping reflections of component 1. The structure was refined using the hklf 5 routine with all reflections of component 1 (including the overlapping ones). The hydrogen atoms were included in idealized positions based on a riding model. All calculations were performed using the SHELXT-2014/4 and SHELXTL-2014/7 [40,41]. The CCDC numbers of **3** and **4** are 1887620 and 1887482, respectively.

## 4. Conclusions

In this study, two tetranuclear manganese complexes, [$Mn_4$(bmsap)$_2$($CH_3CO_2$)$_3$($CH_3O$)] (**3**) and [$Mn_4$(cmsap)$_2$($CH_3CO_2$)$_3$($CH_3O$)] (**4**), were prepared in a satisfactory yield as well as characterized. As we expected, the bromo- and chloro-substituent groups of the present ligands did not affect the tetranuclear features and the crystal structures of these complexes revealed that the present Schiff-base ligands also work with three kinds of acetato ligands (bidentate, syn–anti-bridging, and syn–syn-bridging), and μ-methoxido ligand to construct a Y-shaped tetranuclear arrangement. The magnetic exchange interactions via the bridging ligands were found to be generally weak and mostly antiferromagnetic.

**Supplementary Materials:** The following are available online at http://www.mdpi.com/2312-7481/5/1/8/s1, Figure S1: Diffused reflectance spectra of [Mn4(bmsap)2(CH3CO2)3(CH3O)] (**3**) (upper) and [Mn4(cmsap)2(CH3CO2)3(CH3O)] (**4**) (lower).

**Author Contributions:** M.M. conceived and designed the experiments, analyzed the data, and wrote the paper; S.K., S.T., and Y.K. performed the experiments; D.Y. performed the crystallographic work; R.M. performed the magnetic measurements and crystallographic work. H.S. performed the magnetic data analysis.

**Funding:** The present work was partially supported by Grant-in-Aid for Scientific Research Nos. 15K05445 and 17K05820 from the Ministry of Education, Culture, Sports, Science and Technology (MEXT, Japan) and the MEXT-Supported Program for the Strategic Research Foundation at Private Universities, 2010–2014.

**Conflicts of Interest:** The authors declare no conflict of interest.

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
