# Peer review of "Synthesis, Crystal Structures, and Magnetic Properties of Mixed-Valent Tetranuclear Complexes with Y-Shaped MnII2MnIII2 Core"

_magnetochemistry, doi:10.3390/magnetochemistry5010008_

Round 1

Reviewer 1 Report

    The manuscript by M. Mikuriya et. alreports the reaction of two pentadentate Schiff-base ligands, 1,3- bis(5-bromo-3-methoxysalicylideneamino)-2-propanol (H3bmsap) and 1,3-bis(5-chloro-3-methoxysalicylideneamino)-2-propanol (H3cmsap) with manganese(II) acetate affording two new tetranuclear mixed-valent manganese complexes, [Mn4(bmsap)2(CH3CO2)3(CH3O)] (3) and [Mn4(cmsap)2(CH3CO2)3(CH3O)] (4), respectively.

    The authors describe the synthesis in detail and full characterization of the species through infrared and diffused reflectance spectroscopies and elemental analysis. The manuscript is generally well written and the experimental section is comprehensive and detailed. Single crystals suitable for X-ray diffraction were obtained and the crystal structures of the mixed-valent manganese compounds are well described. Moreover, magnetic susceptibility measurements have been done in order to study the magnetic coupling between the four manganese atoms with different oxidation state. Interestingly, the experimental data have been fitted considering six different exchange coupling parameters (J). The calculated curves match well the magnetic data.  

    Although the authors recently published two very similar compounds, they include a more exhaustive study on the magnetic properties which I think is the most interesting feature concerning this type of coordination compounds. Overall, I think the work is well done and I support its publication, nevertheless I have only a couple of minor comments:

These two tetranuclear compounds show almost the same structure and magnetic properties than the previously reported specie [Mn4(msap)2(CH3CO2)3(CH3O)(H2O)]H2O (1) [DOI: 10.1007/s11696-017-0305-6]. In the present work the authors use the bromo- and chloro derivatives of the H3msap ligand for the synthesis of manganese complexes. Could they conclude with any information about the influence of such groups on the final properties? Are the results in agreement with those they expected when introducing chloro- and bromo- groups to the H3msap ligand?

Page 2, line 68 “finger print” -> “fingerprint”

Author Response

The manuscript by ··· These two tetranuclear compounds show almost the same structure and magnetic properties than the previously reported species [Mn4(msap)2(CH3CO2)3(CH3O)(H2O)]H2O (1) [DOI: 10.1007/s11696-017-0305-6].

Thank you very much for understanding our study.

In the present work the authors use the bromo- and chloro derivatives of the H3masp ligand for the synthesis of manganese complexes.  Could they conclude with any information about the influence of such groups on the final properties?  Are the results in agreement with those they expected when introducing chloro- and bromo- groups to the H3msap ligand?

Thank you for the suggestion.  By using the bromo- and chloro-derivatives, we obtained almost similar tetranuclear compounds to the recently reported compounds as we expected.  We added more comments in the introduction and conclusion.

Page 2, line 68   “finger print”  —>  “finger print” 

Thank you.  We corrected this.

Reviewer 2 Report

The authors present their results on tetranuclear Mn(II)2Mn(III)2 complexes in a clear manner and analyse their magnetic data using PHI using a multi-J model.

Did the authors measure M vs H isotherm data at low temperatures; to try and simultaneously fit the data to the model used for chiT fits?

Did the authors disperse the samples in a polymer or Vaseline to prevent torquing anomalies in Mn(III) systems?

ref 27 spelling Tiekink; check all names in references

Fig 3 put in esi

Table 4   errors?

The Introduction must mention the main focus in recent years on "butterfly" Mn(II)2Mn(III)2 clusters that show ferromagnetic coupling and SMM behaviour. References are given but no specific mention. While the present compounds probably will not show slow magnetisation reversal - did the authors measure ac susceptibilities?

In summary, a reasonable paper, just acceptable.

Author Response

The authors present ··· a multi-J model.

Did the authors measure M vs H isotherm data at low temperature; try and simultaneously fit the data to the model used for chiT fits?

Thank you for valuable suggestion.  We did not measure M vs H isotherm at low temperature because of the limit of our machine time of SQUID.  According to this suggestion, we are planning to measure such properties in the next step of our study together with new compounds.

Did the authors disperse the samples in a polymer or Vaseline to prevent torqueing anomalies in Mn(III) systems? 

Thank you for valuable suggestion.  We did not disperse the samples in a polymer nor Vaseline, because our compounds are mostly antiferromagnetic and we did not observed such phenomenon in our cases.  Cinsidering this advice, we are planning to measure in a such way to make sure of this in the next step of our study together with new compounds.

Ref 27 spelling Tiekinik; check all name in references

Thank you for pointing out this.  We checked and corrected all of the references.

Fig 3 put in esi

We put Figure 3 as Figure 1S in esi.

Table 4 errors?

Unfortunately, the Kambe vector coupling method was not applicable to the magnetic analysis of the present tetranuclear system.  At present, only PHI program could afford a reasonable fitting magnetic data.  Table 4 does not contain the estimated errors for the J values, because it is almost impossible to estimate the errors by using PHI program for the present multi J model case.  

The introduction must mention the main focus in recent years on “butterfly” Mn(II)2Mn(III)2 clusters that show ferromagnetic coupling and SMM behavior. References are given but no specific mention.  While the present compounds probably will not show slow magnetisation reversal-did the authors measure ac susceptibilities?

According to this suggestion, we added comments and some references on “butterfly” Mn(II)2Mn(II)2 clusters.  As pointed out, the present compounds do not seem to show such slow magnetization and we did not have enough machine time to do such studies.  We would like to do this kind of study for in near future work.